# The DARC Side of Inflamm-Aging: Duffy Antigen Receptor for Chemokines (DARC/ACKR1) as a Potential Biomarker of Aging, Immunosenescence, and Breast Oncogenesis among High-Risk Subpopulations

**DOI:** 10.3390/cells11233818

**Published:** 2022-11-29

**Authors:** Nikita Jinna, Padmashree Rida, Tianyi Su, Zhihong Gong, Song Yao, Mark LaBarge, Rama Natarajan, Tijana Jovanovic-Talisman, Christine Ambrosone, Victoria Seewaldt

**Affiliations:** 1Department of Population Science, City of Hope Comprehensive Cancer Center, Duarte, CA 91010, USA; 2Department of Science, Rowland Hall, Salt Lake City, UT 84102, USA; 3Department of Cancer Prevention and Control, Roswell Park Comprehensive Cancer Center, Buffalo, NY 14263, USA; 4Department of Diabetes Complications and Metabolism, City of Hope Comprehensive Cancer Center, Duarte, CA 91010, USA; 5Department of Molecular Medicine, City of Hope Comprehensive Cancer Center, Duarte, CA 91010, USA

**Keywords:** duffy antigen receptor for chemokines, duffy null allele, inflamm-aging, immunosenescence, oncogenesis, high-risk, breast cancer

## Abstract

The proclivity of certain pre-malignant and pre-invasive breast lesions to progress while others do not continues to perplex clinicians. Clinicians remain at a crossroads with effectively managing the high-risk patient subpopulation owing to the paucity of biomarkers that can adequately risk-stratify and inform clinical decisions that circumvent unnecessary administration of cytotoxic and invasive treatments. The immune system mounts the most important line of defense against tumorigenesis and progression. Unfortunately, this defense declines or “ages” over time—a phenomenon known as immunosenescence. This results in “inflamm-aging” or the excessive infiltration of pro-inflammatory chemokines, which alters the leukocyte composition of the tissue microenvironment, and concomitant immunoediting of these leukocytes to diminish their antitumor immune functions. Collectively, these effects can foster the sequelae of neoplastic transformation and progression. The erythrocyte cell antigen, Duffy antigen receptor for chemokines(DARC/ACKR1), binds and internalizes chemokines to maintain homeostatic levels and modulate leukocyte trafficking. A negative DARC status is highly prevalent among subpopulations of West African genetic ancestry, who are at higher risk of developing breast cancer and disease progression at a younger age. However, the role of DARC in accelerated inflamm-aging and malignant transformation remains underexplored. Herein, we review compelling evidence suggesting that DARC may be protective against inflamm-aging and, therefore, reduce the risk of a high-risk lesion progressing to malignancy. We also discuss evidence supporting that immunotherapeutic intervention—based on DARC status—among high-risk subpopulations may evade malignant transformation and progression. A closer look into this unique role of DARC could glean deeper insight into the immune response profile of individual high-risk patients and their predisposition to progress as well as guide the administration of more “cyto-friendly” immunotherapeutic intervention to potentially “turn back the clock” on inflamm-aging-mediated oncogenesis and progression.

## 1. Introduction

Breast cancer is the second largest contributor to cancer-related deaths among American women [1]. Roughly one in eight American women will be diagnosed with breast cancer [2]. Under the age of 40, Black/African-American (AA) women experience significantly higher incidence rates of breast cancer than White/European-American (EA) women and notably higher mortality rates from breast cancer at any age [2].

The immune system, perhaps, mounts one of the most important first lines of defense against breast oncogenesis and progression [3]. This line of defense includes innate and adaptive immune responses, which release critical immune cells to launch an antitumor attack [4]. However, as we age, so does our immune system, which increases susceptibility to malignant transformation and progression. Thus, the risk of breast oncogenesis rises with increasing age [5]. This decline in immune system function as a function of time is often referred to as immunosenescence, which results in “inflamm-aging”, or remodeling of the inflammatory and anti-inflammatory networks [6]. Immunosenescence and inflamm-aging pose a major threat to the ability of tissues to thwart neoplastic transformation and progression via secreting protumorigenic chemokines, increasing infiltration of immunosuppressive cells, and expansion of age-remodeled leukocytes that have diminished in their antitumor immune functions into the tissue microenvironment [7,8,9,10,11]. Researchers have been investigating the molecular mechanisms underlying the predisposition of certain cells to undergo accelerated inflamm-aging for improved disease risk-prediction and therapeutic intervention (Table 1). However, there are currently no robust immune biomarkers available in the clinic that can estimate the propensity of a high-risk lesion more likely to undergo accelerated immunosenescence and inflamm-aging, potentially resulting in malignant transformation or progression.

The Duffy Antigen Receptor for Chemokines (DARC/ACKR1) is a crucial decoy receptor expressed on erythrocytes, endothelial cells, and lymphoblast cells [12]. DARC binds inflammatory chemokines to internalize them for lysosomal degradation [13]. Chemokines, which regulate leukocyte trafficking, play a key role in dictating the immune cellular composition of the tissue microenvironment for proper leukocyte recruitment and antitumor immunity [14,15]. Thus, a lack of homeostatic chemokine levels has been linked to cancer development and progression [14]. Therefore, DARC is critical for counteracting protumoral immunity by preventing excess tissue infiltration of pro-tumorigenic chemokines and immunosuppressive cells.

A mutation in the DARC gene promoter—at the binding site of the GATA1 erythroid transcription factor—blocks erythrocytic DARC expression. This DARC promoter mutation is known as the “duffy-null allele” [13,16,17]. Individuals, who harbor this Duffy-null allele, are less protected from excessive infiltration of pro-inflammatory chemokines and, therefore, evade deregulated leukocyte trafficking into the tissue microenvironment, which can interfere with antitumoral immunity and accelerate carcinogenesis. The vast majority of the West African population harbors this mutation at a 100% fixed allelic frequency [18]. Owing to their predominant West African genetic ancestry, the Black/AA subpopulation harbors the Duffy-null allele at an over 70% allelic frequency [19,20]. This high Duffy-null allele frequency observed among subpopulations of West African genetic ancestry has been suggested to underlie, at least in part, the poorer immune response profile reported among Black/AA compared to White/EA breast cancer patients [21,22,23].

Immunosenescence and inflamm-aging ultimately result in increased infiltration of protumorigenic and proangiogenic chemokines that recruit age-remolded leukocytes into the tissue microenvironment, which fosters tumor development. Since DARC is a master regulator of chemokines and sequesters these protumoral chemokines for degradation, DARC may play a critical role in evading immunosenescence and inflamm-aging via limiting the load of protumorigenic chemokines that enter into the tissue microenvironment and facilitate tumor progression. These events may be evident by the fact that Black/AA individuals, who harbor a higher frequency of the Duffy null allele, exhibit (1) greater lymphocytic tissue infiltration, (2) higher levels of immunosenescence-secreted protumorigenic and angiogenic chemokines, and a (3) greater prevalence of protumoral immune subsets compared to White/EA individuals [21,24,25,26,27]. Furthermore, relative to White/EA individuals, Black/AA individuals develop breast cancer at a notably younger age and are more likely to progress to advanced-stage disease [2]. Collectively, this body of evidence supports the hypothesis that Black/AA individuals may be undergoing accelerated inflamm-aging compared to other ethnic groups as a result of their higher prevalence of a negative DARC status. Hence, DARC status may be highly indicative of the risk for accelerated inflamm-aging and carcinogenesis among high-risk subpopulations.

In this article, we dissect evidence suggesting that DARC may be playing a critical role in evading or delaying inflamm-aging and, subsequently, neoplastic transformation and progression. We propose that individuals with high DARC expression levels may be protected from the protumoral impact of immunosenescence and inflamm-aging, and in contrast, individuals with low DARC levels are at greater risk for immunosenescence, inflamm-aging, and cancer development. Our discussion may encourage increased investigation into the interplay between DARC levels, inflamm-aging, and carcinogenesis for improved risk-prediction and clinical management of high-risk patient subpopulations. This dialogue may also provide a strong impetus for exploiting early-stage immunotherapeutic intervention or administration of aging-prevention drugs, based on DARC status, to derail the progression of high-risk lesions.

**Table 1 cells-11-03818-t001:** Current inflamm-aging-related biomarkers and pathways.

Biomarker	Mechanism	Ref.
Immune cell markers		
NK↑CD14^+^CD56^dim^/↓CD14^+^CD56^bright^	Accumulation of immature NK cells	[28,29]
NK↓NKP30, NKP46, DNAM-1 and ↑KIR, NKG2C	Decrease in activating receptors and increase in inhibitory receptors on NK cells	[29,30,31,32]
Monocyte↑CD14^+^(low)CD16^+^ and CD14^++^(high)CD16^+^/↓CD14^+^(low)CD16^−^	Increase in immature monocytes	[33]
Macrophage↓CD62L and TLR1/4/↑CD11b and TLR5	Decrease in activating receptors and increase in inhibitory receptors on macrophages	[34]
Macrophage M1→M2	Macrophage phenotype switch to proinflammatory	[35,36]
CD8^+^ naïve T cells	Decline in naïve T cells to reduce thymus T cell output	[37,38]
T cell↓CD27, CD28/↑KLRG-1, PD-1, CTLA-4, Tim-3, Tigit, CD57	Downregulation of costimulatory molecules and upregulation of inhibitory molecules on T cells	[39,40]
B cell↓CD19^+^	Downregulation in CD19^+^ B cells to impair B cell function	[41]
*Cytokines*		
↓IFN-γ, granzyme B, perforin	Reduction in dendritic cell and cytotoxic T cell activity	[42,43,44,45,46]
↑IL-10, TGF-β, VEGF, indoleamine-2,3-dioxygenase	Increase in immunosuppressive cytokines	[8,47]
↑IL-6, IL-1, TNF-α, CRP	Increase in proinflammatory cytokines	[8,48,49]
↓IL-7	Reduced activation of T cells	[11]
*miRNAs*		
MiR-9, miRNA-17, miR-10a, miR-19a/b, miR-20, miRNA-21, miR-29a, miR-125a/b, miR-126, and miR-146a, miR-155, miR-181a/b, miR-187, miR-195, miR-199, miR-223, miR-517a/c, Let-7, Let-7i	Modulate signaling pathways implicated in inflamm-aging including NF-κB, mTOR, sirtuins, and TGF-β	[50,51]
*Cell signaling pathways*		
NF-κB	A nuclear transcription factor that serves as the primary molecular switch for inflammatory pathways	[52]
mTOR	Activates NF-κB -mediated regulation of inflamm-aging	[53]
RIG-1	Interacts with increased IL-6 and IL-8 levels in senescent cells and upregulates IL-6 expression	[54]
Notch	Induces senescence of endothelial cells	[55]
Sirtuin	Interacts with NF-κB to suppress its proinflammatory activity	[56]
TGF-β	Anti-inflammatory cytokine that deactivates macrophages to maintain immune homeostasis	[56]
Ras	Upregulates expression of proinflammatory cytokines	[57]

Abbreviations: **↑**, upregulation; **↓**, downregulation; NK, natural killer cells; CD, cluster of differentiation; DNAM-1, DNAX accessory molecule; KIR, killer cell Ig-like receptors; TLR, toll-like receptor; KLRG1, killer cell lectin-like receptor G1; PD-1, programmed cell death protein 1; CTLA-4, cytotoxic T-lymphocyte associated protein 4; Tim-3, T cell immunoglobulin, and mucin-domain containing-3; Tigit, T-cell immunoglobulin, and ITIM domain; IFN, interferon; IL, interleukin; TGF, transforming growth factor; VEGF, vascular endothelial growth factor; TNF, tumor necrosis factor; CRP, c-reactive protein; MiRNA, microRNA; NF-κB, nuclear factor kappa B; mTOR, mammalian target of rapamycin; RIG-1, retinoic acid-inducible gene I.

## 2. Oncogenesis: Cellular Aging, Senescence, and the Senescence-Associated Secretory Phenotype

The risk of developing malignant tumors increases with age; thus, cancer is typically defined as an age-related disease [9,58,59]. Cellular aging or senescence has been suspected to be the cornerstone of malignant transformation for decades. In the 1960′s, Hayflick and Moorhead discovered that human diploid fibroblasts in culture divide a maximum number of times (~50–60) before halting cellular growth irreversibly, also known as the “Hayflick limit” [60]. This phenomenon, known as replicative senescence, occurs as a physiological response as cells age to prevent cancer-causing genomic instability and accumulation of DNA damage [60,61]. Thus, cellular senescence is a physiological tumor-suppressive mechanism that prevents pre-neoplastic cells from progressing into malignant tumors, and the ability of these cells to bypass senescence barriers can result in oncogenic transformation [62,63,64,65]. Conversely, mounting evidence now suggests that the same senescent cells can induce changes in the tissue microenvironment that actually fosters tumorigenesis. The most well-studied mechanism for this senescent cell-mediated neoplastic transformation is the senescence-associated secretory phenotype (SASP).

The SASP is a key feature of senescent cells that has been suggested to facilitate the initiation of cancer and tumor progression via secreting a myriad of molecules, proteins and growth factors that can induce alterations in the tissue microenvironment. Specifically, the SASP can stimulate cell proliferation via secreting growth-related oncoproteins such as b-raf (BRAF), epidermal and insulin-like growth factor (EGF and IGF) receptor ligands such as amphiregulin and IGF-1, and vascular endothelial growth factor (VEGF) to stimulate new blood vessel formation [7,66,67]. The SASP can also stimulate WNT signaling, which can drive stem cell proliferation and differentiation [67]. Furthermore, the SASP can secrete factors that induce epithelial-to-mesenchymal transition, such as matrix metalloproteinases (MMPs), serine proteases, and inhibitors, including urokinase- or tissue-type plasminogen activators (uPA or tPA, respectively), uPA receptor (uPAR), and plasmogen activator inhibitor 1 and 2 (PAI-1 and -2) [68,69,70,71,72,73]. Fibronectin production is another mechanism that the SASP exploits to increase cell adhesion, survival, growth, and migration [74,75]. Additionally, the SASP can secrete molecules, such as reactive oxygen species, and transported ions, such as nitric oxide, that modulate cellular phenotypes to accelerate aging and cancer progression [76,77,78,79,80,81].

Perhaps, the most critical mechanism that the SASP employs to promote malignant transformation and tumor progression is by direct or indirect promotion of inflammation via secreting interleukins (ILs) as well as pro-inflammatory cytokines and chemokines. These inflammatory factors include IL-1, IL-6, IL-7, IL-8, IL-11, IL-15, monocyte chemoattractant proteins, macrophage inflammatory proteins, and inflammatory regulators such as granulocyte-macrophage colony-stimulating factor (GM-CSF) [7,67,82]. IL-6 is perhaps the most critical proinflammatory cytokine secreted by the SASP; IL-6 directly activates neighboring cells in the tissue microenvironment that express the IL-6 receptor (IL-6R) to increase inflammatory activity that can promote oncogenic transformation [7]. IL-1 secretion by the SASP can act on surrounding cells expressing IL-1 receptor (IL-1R) and toll-like receptors to stimulate nuclear factor kappa B and activate protein 1 oncogenic signaling pathways [83,84]. Most senescent cells also overexpress the chemokine CXCL and CCL family members [7]. The CXCL family members released by the SASP include IL-8 (CXCL-8), chemokine growth-regulated protein alpha (GRO-α/CXCL-1), GRO-β/CXCL-2), GCP-2/CXCL-6, ENA-78/CXCL-5, as well as CXCR2(IL-8RB)-binding chemokines. The CCL family members secreted by the SASP include MCP-1,-2,-3,-4 (CCL-2,-8,-7,-13), HCC-4 (CCL-16), eotaxin-3 (CCL-26), macrophage inflammatory protein (MIP)-3α, -1α (CCL-20,-3), and I-309 (CCL-1). Dysregulation or chronic expression of these chemokines has been implicated in tissue damage, angiogenesis, and tumorigenesis [85,86]. Specifically, CXCL-1, IL-6, and IL-8 exhibit autocrine growth effects via stimulating proliferation of the cells they are produced in, and EGF, IL-1, and VEGF have been shown to display paracrine growth effects via modulating the tissue microenvironment to support tumor growth and progression [87]. CXCL-1 has also been suggested to be a diagnostic biomarker of aging and cancer [87]. Furthermore, the SASP has been shown to suppress the antitumor immune response and promote tumor metastasis via secreting cytokines and chemokines that recruit tumor-infiltrating myeloid-derived suppressor cells (MDSCs) [88,89]. These recruited MDSCs to block CD8^+^ T cell response via releasing IL-1 receptor antagonists that interfere with IL-1 signaling in tumor cells [90].

Collectively, these changes in the tissue microenvironment have been suggested to accelerate immune system aging or immunosenescence [91]. The factors secreted by the SASP into the microenvironment can induce senescence of surrounding immune cells and drastically deregulate their function [9]. Furthermore, alterations in the tissue microenvironment during aging has been linked to metastasis in elderly cancer patients [92,93]. Thus, the SASP is the cornerstone of immunosenescence-mediated tumor development and progression [37,94].

## 3. Inflamm-Aging: Illuminating Immunosenescence

The immune system plays a critical role in thwarting malignant transformation and tumor progression via immune surveillance and mounting an antitumor response. Dysregulation of the immune system can mount a defective antitumor immune response and result in tumor initiation and progression [95]. Immunosenescence is a natural progression of immune cells in the tissue microenvironment that accompanies an increase in age and results in a decline in immune system function. It has been characterized by a decline in innate and adaptive immunity as well as infection resistance but increased risk for autoimmunity [10,11]. In fact, studies show that the innate and adaptive antitumor immune systems differ between young and elderly individuals, with younger individuals displaying stronger and more effective antitumor immunity [9]. Inflamm-aging occurs as a result of these age-related changes and is considered to be low-grade chronic inflammation, in which the anti-inflammatory response is overwhelmed by the pro-inflammatory response [8]. This dysfunctional immune system can lead to a weak antitumor immune attack and potentially result in tumorigenesis and progression. Although limited, many groups have begun unraveling the mechanisms of how the immunosenescence of immune cell mediators in the tissue microenvironment predisposes cells to malignant transformation and progression.

### 3.1. Innate Immunosenescence

Natural killer (NK) cells dominate the antitumor innate immune response via immune surveillance. However, aging can hinder their activity, which can lead to oncogenesis. The levels of mature NK cells were observed to significantly decrease in all lymphoid organs in aged mice [28]. Increased accumulation of immature CD14^+^CD56^dim^ and mature CD14^+^CD56^bright^ indicates aging and remodeling or immunoediting of mature NK cell subsets as well as an increased risk for disease progression [28,29,30,96]. Furthermore, the expression of activating receptors (i.e., NKP30, NKP46, and DNAM-1) on NK cells, which allows them to recognize and lyse tumors, is often decreased, while the expression of inhibitory receptors (i.e., KIR, NKG2C) is often increased on NK cells during immunosenescence [29,30,31,32]. Collectively, this remodeling of the NK cell profile is often characterized by a reduced capacity to respond to cytokines, which can result in dendritic cell inactivation and reduced interaction with macrophages [35,97].

In addition to age-related changes in NK cells, monocytes and macrophages also undergo phenotypic remodeling during immunosenescence. This immunoediting includes an increase in CD14^+^(low)CD16^+^ and CD14^++^(high)CD16^+^ populations and a decrease in CD14^+^(low)CD16^−^ monocytes [33]. A decrease in the expression of CD62L and TLR1/4 and an increase in CD11b and TLR5 expression on macrophages were also observed, which can facilitate neoplastic progression [34]. In addition, macrophages exhibit decreased phagocytic activity as well as a switch from an M1 to and M2 proinflammatory phenotype during aging [36,98]. The ability of neutrophils to phagocytose pathogens and recruit chemokines also wanes, along with them being more prone to apoptosis with age [99,100,101]. These age-related changes in macrophages and neutrophils can result in chronic low-grade inflammation and suppression of the immune system, which has been linked to carcinogenesis [99]. It has also been discovered that circulating levels of dendritic cells significantly decline with age [102]. Additionally, the antigen presentation, endocytic, and interferon-gamma (IFN-γ) production functions of dendritic cells are also weakened during immunosenescence, including the ability to prime CD8^+^ and CD4^+^ T cells [42,43,44,103]. MDSCs, which impair T cell function and tumor cell clearance, increase in the bone marrow, blood, and spleen of aged mice bearing tumors [89]. MDSCs can also induce changes in the tissue microenvironment during aging by secreting TGF-β and IL-10 [47].

### 3.2. Adaptive Immunosenescence

A critical feature of adaptive immunosenescence that elevates the risk of neoplastic transformation is the decline in CD8^+^ naïve T cells and, therefore, T cell output as a result of the thymus gland degenerating with age [37,104]. This decline in naïve T cells can reduce T cell antigen receptor diversity and disrupt T cell homeostasis [38,105,106,107]. Thus, reduction in these cytotoxic T cells can lead to a weaker antitumor immune response and, therefore, increased susceptibility to tumorigenesis. The cytotoxic activity of these T cells can also wane during immunosenescence, including a significant reduction in the expression of the functional molecules involved in this cytotoxic response, such as IFN-γ, granzyme B, and perforin [45,46]. Conversely, naïve memory T cells have been discovered to accumulate with age [108]. Senescent T cells can also release pro-inflammatory cytokines such as IL-6 and TNF-α [48,49]. Furthermore, costimulatory molecules of T cells such as CD27 and CD28 have been shown to be downregulated, whereas the expression of inhibitory receptors such as killer cell lectin-like receptor subfamily G (KLRG-1), programmed cell death protein 1 (PD-1), cytotoxic T-lymphocyte-associated protein 4 (CTLA-4), Tim-3, Tigit, and CD57 have been uncovered to be upregulated on T cells during immunosenescence [39,40]. This upregulation in immune checkpoint-related molecules can increase the risk of tumor development and progression. In addition, these senescent T cells can exhibit decreased replicative capacity and reduced survival upon T cell receptor activation [39]. T cell senescence is also often accompanied by an increase in levels of tumor-associated macrophages (TAMs) and regulatory T cells (Tregs) that can suppress the antitumor immune response [109,110,111,112,113]. Downregulation in CD39 expression during aging has been linked to the reduction in CD4^+^ T cell levels [114]. Levels of the PD-1^+^ memory CD4^+^ T cells gradually increase with age and are the predominant phenotype in aging mice [115].

Similar to T cells, B cells, which produce antibodies to mount an antitumor response, undergo remodeling during aging. The bone marrow niche becomes skewed toward generating myeloid cells as opposed to lymphocytes, and the levels of circulating naive B cells significantly decline during aging [116,117]. Mature B cell subsets are redistributed, and their activation is also weakened during the aging process [118,119]. B cell receptor repertoire diversity is lost during aging, which may reflect the expansion of memory B cell clones [120]. These age-related changes may alter antibody specificity and increase auto-antibodies. The proportion of CD19^+^ B cells in the peripheral blood decrease with increasing age along with impairment of their B cell function [41]. This event is often accompanied by a reduction in autoimmune regulator (AIRE) expression and autoantigen genes in thymic B cells [121]. In addition, mature spleen B cell turnover often declines with age [122,123].

A key event that occurs during immunosenescence and increases vulnerability to oncogenesis is a disruption in the IL-7/IL-7R signaling network [124]. The IL-7/IL-7R pathway plays a significant role in the development and homeostatic regulation of T, B, and NK cells. In fact, as the thymus ages, IL-7 production decreases, which has been associated with compromised activation of T cells [11]. Hence, dysregulation in IL-7/IL-7R signaling can interfere with proper immune function and the successful mounting of an antitumor attack.

### 3.3. Inflamm-Aging

Collectively, this age-related decline in innate (i.e., reduced NK, DC, monocyte/macrophage, and neutrophil functions but increased MDSC functions) and adaptive (i.e., reduced levels of naïve T cells but increased levels of memory T cells) immunity, ultimately results in low-grade chronic inflammation or inflamm-aging. This inflamm-aging further alters the immune response by producing more pro-inflammatory cytokines such as IL-1, IL-6, and TNF-α as well as generating more immunosuppressive cells and cytokines such as indoleamine-2,3-dioxygenase, TGF-β, IL-10, VEGF, and PD-1 [8]. These resulting changes further increase susceptibility for tumor development and progression.

## 4. A Light in the DARC: Gaining Back Time

### 4.1. Chemokines: Small Proteins, Big Roles

Chemokines are small, secreted proteins whose “roles and responsibilities” portfolios are vast and diverse [125]. Chemokines display cytokine-like activities and serve as leukocyte chemo-attractants that regulate the migration of appropriate receptor-bearing cells to sites of inflammation [126,127]. They are categorized on the basis of the number and position of conserved cysteine residues in two major (CXC and CC chemokines) and two minor (C and CX3C chemokines) subsets and according to the stimuli that provoke their production into homeostatic (i.e., produced constitutively and regulate homeostatic trafficking of leukocytes and lymphocyte recirculation) and inflammatory (i.e., induced in response to inflammatory or immunological stimuli, and direct leukocytes to inflamed peripheral tissues) molecules [128,129]. Chemokines engender extracellular patterns of chemotactic or haptotactic gradients, depending on whether they are in solution or bound to extracellular matrix components, respectively. Furthermore, chemokines are immobilized and presented on endothelial cell surfaces; they may then cause rolling leukocytes that harbor cognate receptors to stop, adhere firmly, and then undergo extravasation. Following this trans-endothelial migration, leukocytes further respond to chemokine gradients by directional migration into specific microdomains. Thus, chemokines are implicated in modulating leukocyte entry into the circulation, extravasation from blood vessels, influencing leukocyte placement in tissue microenvironmental contexts, and their function in innate and acquired immunity [128].

It is no wonder that chemokines have emerged as powerful regulators in the pathophysiology of cancer. Preexisting chronic inflammatory conditions can increase the risk of cancer developing, and conversely, oncogenic mutations can lead to the establishment of an inflammatory tumor microenvironment (TME), a hallmark of cancer [130,131]. As key mediators of chronic inflammation that may also be transcriptionally induced by oncogenes and transcription factors deregulated in the pathogenesis of cancer, chemokines are culpable in both these aspects [131]. Their altered expression can shape the evolution of the neoplastic process in profound ways as they dictate leukocyte recruitment and activation, angiogenesis, cancer cell proliferation, and metastasis in all stages of the disease. Chemokines can be produced by tumor cells themselves, by cancer-associated fibroblasts, and by infiltrating leukocytes. When inflammatory leukocytes accumulate at a tissue site, they generate an environment that promotes early tumor development, possibly through the production of cytokines, proteases, and angiogenic factors. Furthermore, the transformation of a preneoplastic lesion to a neoplastic state, growth of tumors larger than 2–3 mm [3], invasion, and metastases all depend on the establishment of a proangiogenic environment, which is produced when the local concentrations of angiogenic factors exceed those of angiostatic ones [132]. In 1995, Strieter et al. showed that CXC-ELR+ chemokines have angiogenic properties; by contrast, CXC-ELR- chemokines have angiostatic properties [133]. Angiogenesis makes available routes for malignant cells to access and enter circulation for subsequent systemic dissemination and is also important in the establishment of these cells at the site of metastasis [134].

The spatial and temporal expression patterns of chemokines are critical determinants of the composition of the TME and the behaviors of various tumor-infiltrating leukocytes. Inflammatory chemokines recruit immature cells of myeloid origin, or MDSCs, to the tumor site [135], as well as more mature cells, such as monocytes and neutrophils, which then differentiate into tumor-associated macrophages (TAMs) or tumor-associated neutrophils (TANs) [136]. Other immune cells, such as lymphocytes, cancer-associated fibroblasts (CAFs), mesenchymal stem cells (MSCs), and endothelial cells, may also be recruited into the TME [137,138,139]. Collectively, these infiltrating cells provide a secondary source of chemokines that may affect tumor growth, cell survival, senescence, angiogenesis, and metastasis. Cancer cells produce chemokines and chemokine receptors which enable them to respond specifically to chemokines in their milieu, thus forming a complex chemokine network. MDSCs are subverted in their function and aid and abet tumor growth by promoting neo-angiogenesis and impeding anti-tumor T-cell responses [140]. Furthermore, some chemokines can lead to the recruitment of Tregs and Th2 lymphocytes that hinder antitumor responses and thus support tumor survival [141,142]. Chemokines also direct tumor cell migration in vivo. The expression of chemokine receptors by tumor cells not only endows them with the ability to metastasize and colonize specific anatomical sites where their cognate chemokine is present but also bolsters their proliferation via the activation of the PI3K-AKT-NF-kB, MEK1/2, and Erk1/2 axes [143,144]. Additionally, by tipping the balance of pro-apoptotic and pro-survival factors, chemokines may favor the survival of malignant cells [145,146]. Thus, the gamut of cancer-related roles chemokines regulate extends beyond immunity and inflammation to the regulation of leukocyte recruitment into the tumor mass, promotion of tumor cell survival, proliferation, and dissemination.

### 4.2. DARC: A Time Decoy

Cytokines and chemokines elicit their effects by binding and signaling via signaling receptor complexes. Cytokine and chemokine activity needs to be tightly regulated for the proper antitumor immune response. Overactive immune signaling can lead to immune cell exhaustion, inflammation-induced tissue damage, and autoimmunity and, thus, be counterproductive in preventing tumor growth and progression [147]. Decoy receptors, also known as silent non-signaling receptors, are often employed by the immune system to counteract and balance out this overactive immune response via serving as a molecular “sink” that traps and internalizes these pro-inflammatory cytokines and chemokines for lysosomal degradation, thereby suppressing their immune activity [83,148,149,150]. Thus, decoy receptors serve a critical role in mounting a successful antitumor immune attack.

DARC is one of the most known critical decoy receptors. DARC is a seven-transmembrane G-protein-coupled receptor that was originally discovered to be expressed on human erythrocytes, or red blood cells, and serves as a portal to allow the malarial parasite, *P. vivax*, to enter and infect [151]. It is also expressed on endothelial cells that line post-capillary venules and has recently been observed on lymphoblast cells [12]. Evidence now supports the primary role of DARC as a depot for a host of pro-malignant and pro-inflammatory chemokines involved in immunosenescence, such as CXCL1, CXCL2, CXCL3, CXCL4, CXCL5, CXCL7, CXCL8/IL-8, CXCL12, CCL2, CCL5, CCL7, CCL11, CCL13, CCL14, and CCL17 [152,153]. Thus, DARC heavily influences and modulates (a) homeostatic levels of chemokines in circulation and (b) leukocyte trafficking, which is critical for forestalling tumorigenesis and progression [154].

A mutation in the promoter of the GATA box in the DARC gene removes DARC expression from erythrocytes, which is known as the Duffy-null allele [13,16,17].” The majority of the West African population harbor this mutation at a 100% fixed allelic frequency and, thus, are resistant to *P. vivax*-induced malaria [18]. The Black/AA subpopulation harbors the Duffy-null allele at an over 70% allelic frequency, owing to their predominate West African genetic ancestry [19,20]. Although more protected from malaria infections, individuals of West African descent are less likely to harbor homeostatic chemokine circulation and leukocyte trafficking owing to their DARC-negative status, which may explain their less effective antitumor immune response compared to individuals of European descent [21,22,23,24]. Thus, this deficiency may underlie the increased incidence of breast tumor growth in Black/AAs at a younger age and has been suggested to underlie the large gap in breast cancer outcomes between Black/AA and White/EA patients [155,156].

### 4.3. Shedding Light on Erythrocytic DARC

The expression of DARC is under tight tissue-specific regulation. The Duffy-null allele does not affect the expression of DARC in non-erythroid tissues, including endothelial cells [149]. Erythrocyte DARC is believed to act as a chemokine buffer; on the one hand, it sequesters cognate chemokines when the latter is present at excessively high levels in the serum [157] and dampens plasma chemokine surges. On the other hand, erythrocytic DARC serves as a chemokine reservoir or depot that helps maintain a homeostatic level of these chemokines in the plasma by discharging them into the circulation when the plasma chemokine concentrations subside [157,158]. Since plasma chemokines likely desensitize circulating leukocytes, buffering by erythrocytic DARC potentially preserves leukocyte sensitivity to pro-inflammatory chemokines. By muting circulating chemokine “noise”, erythrocytic DARC may also enhance leukocyte emigration in response to the chemokine signals.

### 4.4. No Longer a DARC Secret: Endothelial DARC Mediates Chemokine Transcytosis

When present on the cell membranes of polarized endothelial cells, DARC mediates abluminal internalization and transcellular transport of cognate chemokines. Endothelial DARC internalizes chemokines on the basolateral cell surface by a micro-pinocytosis-like process, and transports them unidirectionally to the apical surface of the endothelial cell, where they are immobilized on the tips of microvilli [159]. In this manner, DARC allows pro-emigratory chemokines produced in the extravascular tissues to be presented to circulating cells, such as leukocytes, which may then be recruited to inflammation sites [157,160,161,162]. Endothelial DARC is, therefore, referred to as a “chemokine interceptor”, or internalizing receptor. Endothelial DARC thus prevents the escape of soluble tissue-derived chemokine molecules into circulation and allows them to associate with the tips of luminal microvilli and stimulate firm adhesion of leukocytes and subsequent extravasation [154]. Additionally, the removal of chemokines from perivascular spaces may be a potential mechanism by which endothelial DARC can negatively regulate chemokine-induced angiogenesis. The pattern of endothelial cells that express high levels of DARC is similar to areas that are key in leukocyte trafficking and extravasation [163,164]. Expression of DARC on endothelial cells additionally leads to the senescence of these cells and subsequent attenuation of angiogenesis [165].

### 4.5. Shot in the DARC: How DARC Stops Circulating Tumor Cells in Their Tracks

Most cancer patients ultimately succumb to metastatic disease. During metastasis, the attachment of cancer cells to the endothelial cells on microvasculature not only determines the physical site of metastasis but also provides the necessary anchorage to facilitate tumor cell extravasation. However, recent evidence indicates that this interaction also serves as a host defense mechanism that thwarts metastasis. The membrane-bound KAI1/Cd82 protein, also known as Kangai1 (Kai1), is a bona fide metastasis suppressor protein in multiple cancer types [166,167,168]. KAI1 is highly expressed in the normal epithelial cells of the prostate, breast, and lung, and its expression is substantially reduced in carcinoma [166]. Recent studies showed that in addition to transcytosing chemokines, endothelial DARC also binds KAI1 in post-capillary venules. Tumor cells dislodged from the primary tumor and expressing KAI1 (a member of the tetraspannin family) attach to endothelial cells in the post-capillary venules and directly interact with DARC. This direct interaction inhibits circulating tumor cell proliferation and induces senescence in the tumor cells by regulating the expression of the transcription factor TBX2 and the cyclin-dependent kinase inhibitor p21 [169]. The tumor cells at the senescent stage (when KAI1 binds DARC) are expected to be cleared swiftly by immune cells in the blood vessels, as it has been reported that senescent tumor cells trigger innate immune responses which target these tumor cells [165]. Such direct physical contact between endothelial and “in transit” tumor cells can control the survival of the disseminating metastatic cells. These lines of evidence position DARC as a promising therapeutic candidate for preventing the onset of metastatic disease or for neutralizing disseminated tumor cells. They also raise the intriguing possibility that DARC expression on tumor cells may limit the proliferation of KAI1-positive angiogenic endothelial cells. Studies have shown that overexpression of DARC in breast cancer cells significantly suppressed spontaneous pulmonary metastases [170]. Therefore, DARC may even function as a metastasis suppressor. In sum, DARC can modify functional chemokine gradients and patterns in tissues via local regulation of chemokine presentation and function.

## 5. Painting a DARC Picture: DARC, Immunosenescence, and Neoplastic Transformation

Chemokines are key modulators of the antitumor inflammatory response by dictating the composition of the tissue microenvironment via leukocyte recruitment and activation, angiogenesis, cell proliferation, and metastasis [14]. Aberrant levels of chemokines have been associated with an exacerbated or defective inflammatory response and suggested to underlie cancer development and progression [171,172]. As previously mentioned, DARC sequesters and degrades these chemokines that are often secreted by the SASP as well as regulates the trafficking of leukocytes that can display age-related changes during immunosenescence. Thus, DARC status may be able to indicate the likelihood of inflamma-aging-induced malignant transformation and progression. Specifically, DARC expression levels may reveal the capacity of clearing pro-tumorigenic and pro-metastatic chemokines secreted by the SASP, reducing infiltration of immunosuppressive cells, and limiting trafficking of age-remodeled leukocytes that have declined in their antitumoral functions into the tissue microenvironment (Figure 1).

DARC can limit levels of chemokines released by the SASP that recruit immunosuppressive cells into the tissue microenvironment that hinder antitumoral immunity, such as TAMs, Tregs, MDSCs, and TANs. Specifically, DARC binds and internalizes the chemokines (i.e., CCL2, CCL5, CCL7, CXCL1, CXCL2, CXCL5, and CXCL8) responsible for recruiting monocytes and neutrophils, which differentiate into TAMs and TANs to exert pro-tumoral effects via suppressing antitumor immune cells [173,174,175,176]. CCL2, in particular, recruits TAMs and MDSCs, which inhibit CD8^+^ T cell activation [177,178,179,180,181]. CCL5 has been associated with breast carcinogenesis and progression via increasing TAM infiltration [182,183]. These TAMs also release the chemokine, CCL22, which recruits Tregs into the tissue microenvironment to suppress T cell-mediated antitumor immunity [184]. DARC can also trap the chemokines (i.e., CCL5) that attract dendritic cells and Tregs to promote tumor growth and proliferation [185,186]. These chemokines can also activate growth signaling pathways, such as phosphoinositide 3-kinase (PI3K)/AKT and extracellular signal-regulated protein kinases 1 and 2 (ERK 1/2), to promote cell proliferation [144,173,187]. Hence, DARC can prevent excessive infiltration of immunosuppressive immune cells and growth signaling activation that can support malignant transformation and progression.

DARC also regulates levels of pro-angiogenic chemokines secreted by the SASP. DARC can internalize chemokines (i.e., CCL2, CCL11, CXCL1, CXCL2, CXCL3, CXCL5, CXCL7, CXCL8) that promote angiogenesis or formation of blood vascular, which can foster tumor growth, as well as promote endothelial cell survival via inhibiting apoptosis [132,188,189]. TANs, which are also recruited by chemokines that DARC binds and sequesters, can also secrete angiogenic factors that enhance angiogenesis [190]. A major ligand for DARC secreted by the SASP, CXCL8, upregulates VEGF expression to also stimulate new vessel formation and the production of additional angiogenic chemokines [189]. Conversely, DARC does not bind angiostatic chemokines that inhibit angiogenesis, such as CXCL10 and CXCL9 [191]. DARC also modulates levels of chemokines involved in tumor progression. Specifically, DARC internalizes CXCL12, which signals via the CXCL12/CXCR4 axis, to promote migration and metastasis of cancer cells [172,192]. Inhibition of the CXCR4/CXCL12 axis has been shown to suppress breast tumor metastasis to the lung [15]. DARC has also been implicated in modulating levels of other pro-tumorigenic inflammatory factors released by the SASP. Specifically, DARC has been linked to regulating IL-1β, IL-6, and TNF-α inflammation levels in mice post-bone fracture [193]. Hence, DARC may be playing a critical role in limiting circulating levels of pro-tumorigenic chemokines secreted by the SASP, which may avert tumorigenesis and progression.

DARC may also be playing a critical role in attenuating the detrimental effects of innate and adaptive immunosenescence. Regarding innate immunity, DARC may be able to internalize chemokines that recruit age-remodeled NK cells, neutrophils, and macrophages that have waned in their antitumor abilities and can foster tumorigenesis. During adaptive immunosenescence, DARC may be clearing chemokines that stimulate migration of aged T cells that have declined in their cytotoxic activity, exhibited a reduction in the expression of their functional molecules (i.e., IFN-γ, perforin), and downregulated in costimulatory molecules (i.e., CD27) but upregulated in coinhibitory molecules (i.e., KLRG-1, PD-1, CTLA-4). These senescent T cells also secrete pro-tumorigenic cytokines such as IL-6 and TNF-α that are also regulated by DARC. T cell senescence can also increase the recruitment of TAMs and Tregs that release immunosuppressive chemokines that DARC can bind and sequester. Thus, DARC expression levels may indicate the capacity to deaccelerate inflamm-aging.

Low DARC expression levels have been linked to the onset of tumorigenesis and progression. Overexpression of DARC in breast cancer cells transfected into mice xenografts led to the in vivo inhibition of tumorigenesis and metastasis via interfering with neovascularity [148]. This inhibition was accompanied by a reduction in CCL2 levels, microvessel density, and MMP-9 expression. Negative DARC expression was also strongly associated with neoangiogenesis as well as lymph node, bone, and hepatic distant metastasis in breast cancer patient tissue [194]. DARC-negative mice displayed enhanced prostate tumor growth along with increased levels of the angiogenic chemokines CXCL8 and CXCL2 [195]. Downregulation of DARC has also been suggested to potentiate colorectal and melanoma tumor development and progression via increased angiogenesis [196,197].

Since immunosenescence occurs as a function of time, age is a well-established risk factor for breast cancer. Women display a significantly higher likelihood of developing breast cancer at an older age [198]. Rates of breast cancer remain low in women under the age of 40 but rapidly increase after age 40 and are the highest among women over the age of 70. Approximately only 4% of women develop breast cancer under the age of 40 in the U.S., according to the American Cancer Society. However, Black/AA women notoriously exhibit higher breast cancer incidence rates at a younger age in comparison to their White/EA counterparts. The median age at diagnosis for White/EA women is 63, whereas, for Black/AA women, the median age is 60 [2]. The median age at death is 70 for White/EA women and 63 for Black/AA women [2]. Under the age of 40, Black/AA women display significantly higher incidence rates and are more likely to die from breast cancer compared to non-Hispanic White/EA women [2,155]. Specifically, incidence and survival disparities between Black/AAs and White/EAs are largest among the young subpopulation but decline with age [2]. These indirect lines of evidence collectively support the notion that Black/AAs are undergoing aging and likely inflamm-aging at a significantly faster rate than their White/EAs counterparts.

Since, as previously mentioned, the Duffy null allele is significantly more prevalent in individuals of West African ancestry and their weaker antitumor immune response has been suggested to underlie their poor prognosis, the lack of DARC expression among Black/AAs could be associated with their faster rates of inflamm-aging and, therefore, higher incidence rates of breast cancer and mortality at a younger age compared to individuals of European ancestry. It was discovered that among invasive tumors in The Cancer Genome Atlas (TCGA), Black/AAs exhibited a significantly higher proportion of DARC-low expression compared to White/EAs, and DARC tumor expression positively correlated with levels of the pro-inflammatory chemokines released by the SASP, CCL2 and CXCL8 [26]. Furthermore, Black/AA, compared to White/EAs breast cancer patients, harbor higher levels of chemokines released by the SASP, such as IL-6, IL-8, VEGF, CCL7, and CCL8, which has been suggested to correlate with lower levels of DARC expression in the Black/AA subpopulation [25]. Moreover, among breast cancer patients, Black/AAs harbor greater levels of immunosuppressive cells recruited by the SASP via cytokines and chemokines, such as TAMs consisting of the protumorigenic M2 phenotype, Tregs, and MDSCs [25]. The Black/AA subpopulation harbor higher expression levels of immunosuppressive markers such as PD-1, PD-L1, and CTLA-4 compared to the White/EA group [199]. In TNBC, Martini and colleagues recently showed that West African genetic ancestry is significantly associated with PD-1 expression [200]. Within the inherently aggressive TNBC subtype, Black/AAs display greater immune infiltration and inflammation compared to non-Black/AA patients [21]. The Women’s Circle of Health Study recently reported that Black/AA patients harbor a significantly higher density of CD8^+^ T cell tumor infiltration but a more exhausted CD8^+^ T cell profile compared to White/EA breast cancer patients [24,27]. Martini et al. recently uncovered that West African genetic ancestry is highly associated with increased immune cell migration and infiltration but simultaneous repression of immune cell activation or naïve cells in TNBC [200]. The group also discovered that shared West African genetic ancestry is highly associated with greater infiltration of CD8^+^ memory T cells and immunosuppressive FOXP3^+^ Tregs. Based upon these indirect lines of evidence, we postulate that the absence of DARC could be contributing to individuals of West African descent being less protected from clearing pro-tumorigenic chemokines, reducing infiltration of immunosuppressive cells, and age-remodeled leukocyte recruitment, and may thus be more susceptible to undergoing neoplastic transformation and progression at a younger age. Thus, the investigation into the potential role of the Duffy null allele and accelerated inflamm-aging in underlying higher incidence rates of breast cancer at a younger age among the Black/AA subpopulation could yield valuable insights into circumventing this major health disparity.

## 6. Turning Back the Clock on Time: Intervention Strategies for Employing DARC as a Biomarker of Breast Inflamm-Aging, Oncogenesis, and Immunotherapy Response among High-Risk Subpopulations

The immune system remains the cornerstone of fighting off tumor growth and progression. Thus, inflamm-aging poses a major threat to the immune system in successfully accomplishing this task. Since approximately one in eight American women will develop breast cancer in their lifetime, and certain subpopulations are at higher risk for developing the disease at an earlier age, such as Black/AAs, certain individuals may be more susceptible to the detrimental effects of immunosenescence than others [155]. Hence, this disparity warrants a closer look to identify high-risk subpopulations that are predisposed to undergoing accelerated inflamm-aging, which could result in neoplastic transformation and progression, as well as a more favorable response to immunotherapy as an early-stage intervention strategy.

Atypical ductal hyperplasia (ADH) is a common abnormal but benign breast lesion that is four- to five-fold more likely to progress to malignancy or ductal carcinoma in situ (DCIS) than a normally benign lesion [201]. It is discovered by clinicians among approximately 5–20% of breast biopsies and classified as a “high-risk” lesion. DCIS encompasses roughly 20–25% of all breast cancer diagnoses and harbors a greater risk of developing into invasive breast cancer [202]. Current clinical management of these pre-malignant and pre-invasive lesions has been challenging with the severe lack of risk-stratifying biomarkers and, therefore, reluctant administration of cytotoxic and invasive treatments such as lumpectomies, mastectomies, radiation, and endocrine therapy [201,202]. Thus, there is an urgent need for robust biomarkers that can identify high-risk subpopulations harboring a greater propensity to progress and most susceptible to targeted therapeutic intervention. Since the immune system plays a critical role in preventing neoplastic transformation and progression, robust biomarkers of accelerated inflamm-aging may be valuable in improving risk-stratification and management of high-risk subpopulations.

We previously discussed that DARC could play a key role in protecting individuals from immunosenescence-mediated malignant transformation and progression. This role may be accomplished via DARC clearing protumorigenic chemokines, reducing infiltration of immunosuppressive cells, and reducing recruitment of age-remodeled leukocytes that have waned in their antitumor immune function into the tissue microenvironment. Thus, DARC may serve as a robust biomarker of inflamm-aging in pre-malignant and pre-invasive patient subpopulations and aid in predicting the likelihood of a high-risk patient progressing to DCIS or invasive breast cancer.

Drug interventions have been shown to slow down the aging process and deter the onset or progression of age-related diseases [203]. Some of these drugs include metformin, an antidiabetic drug, and nicotinamide mononucleotide (NMN), which increases NAD+ levels in the body; these compounds have been shown to prevent aging and age-related diseases in preclinical models [204,205]. Inhibiting the mTOR pathway via the anticancer drug rapamycin is perhaps one of the most effective strategies in slowing or reversing age-related changes in preclinical models [206,207]. One of the critical mechanisms rapamycin has demonstrated to achieve this prolonged lifespan is via reversing the increase in SASP among the senescent cell population [208,209,210]. Hence, administering these aging-prevention drugs to high-risk populations at a higher risk of undergoing accelerated inflamm-aging, or exhibiting a low DARC status, could potentially reduce or slow down inflamm-aging, therefore, derailing neoplastic transformation or progression. Immunotherapeutic strategies currently in clinical trials and or approved for clinical practice may also circumvent inflamm-aging in high-risk patient subpopulations. These strategies may include adoptive T cell therapy to replenish levels of cytotoxic T cells, chemokine inhibitors (i.e., CCL2/CCR2 axis inhibitors such as PF-04136309 and CCX872) to reduce excessive infiltration of pro-inflammatory chemokines to restore homeostatic levels, and immune checkpoint therapies (i.e., PD-L1, CTLA-4) to counteract the immunosuppressive functions of the SASP and age-remodeled leukocytes [14,211,212]. Essentially these cutting-edge immunotherapies may be able to “fine tune” the immune system of a high-risk patient, who may be at risk for accelerated inflamm-aging, to mount a stronger immune defense against malignant growth and progression.

Precise evaluation of DARC expression levels in high-risk patient samples will be critical in implementing DARC as a risk-stratifying biomarker for progression and immunotherapeutic response. DARC is expressed on endothelial cells even in individuals that are negative for DARC expression in erythrocytes [149]. Hence, accurate evaluation of DARC levels for refining clinical decisions for high-risk patients will require understanding and investigation into how the interplay between erythoid and endothelial expression influences chemokine levels and leukocyte recruitment. Furthermore, accumulating evidence suggests that endothelial DARC function may be more complex than erythoid DARC, suggesting that further investigation into these differences will be critical to exploiting DARC successfully for clinical management [195]. Jenkins and colleagues demonstrated the feasibility of integrating immunohistochemistry (IHC) DARC expression into the clinic to glean insight into the levels of circulating chemokines and immune cell infiltration into the tissue microenvironment via quantitating tumor-specific DARC epithelial cell expression [26]. This practical methodology could potentially reveal to clinicians the immune response profile of an individual high-risk patient for optimal prediction of progression and immunotherapeutic intervention response. Understanding the prognostic and predictive role of DARC expression on each cell type (erythoid, endothelial, and epithelial) and how it pertains to an individual’s immune response profile will be critical to implementing IHC-based DARC in the clinic for high-risk patient management.

We also propose that for research purposes, the adoption of more comprehensive three-dimensional spatial phenotyping approaches, in combination with consideration of the host genotype, may yield rich insights and help us better understand how DARC truly shapes the composition and architecture of the TME in each individual patient, and enable more rational and precise prediction of disease progression. As discussed earlier, a complex network of interactions between chemokines and their receptors sculpts the TME immune cell landscape and the patterns and extent of neovascularization. For example, the chemotactic gradients and immune cell populations, distributions, interactions, and cell states in DARC-negative tumors, would differ substantially from those in DARC-positive ones and need to be assessed in three dimensions across the tumor space, compared across and between microdomains and cell neighborhoods, with extensive quantification of cell types and mapping of vascularization patterns. Such careful consideration of spatial context and development of datasets by multiplexing several biomarkers and deploying high-resolution multiplex immunofluorescence and image analysis approaches is more likely to help us derive valid and biologically meaningful signatures of disease development and progression. Only then can we perhaps discern and reliably predict which tumors are more likely to progress or to respond to specific therapies.

Immunosenescence and inflamm-aging pose a major threat to patients at higher risk for developing breast malignancy and progressing to invasive breast cancer. Thus, a closer look into the molecular mechanisms driving accelerated inflamm-aging in some patients and not others could unlock key biomarkers that may significantly improve clinical risk-stratification and management of high-risk subpopulations. Characterization of circulating senescent cells for appropriate immunotherapeutic intervention can be time-consuming and complicated via exploiting techniques such as multicolor flow cytometry and single-cell RNA sequencing [213]. DARC status represents a promising single-biomarker IHC-based approach that may be able to inform clinicians of the propensity of a high-risk patient progressing based on their unique immune response profile and guide potential immunotherapeutic intervention to reduce their risk of progression. Preclinical investigation of the role of erythoid, endothelial, and epithelial DARC expression in human mammary epithelial cell systems and in patient-derived in vivo organoid models that reflect neoplastic progression will be pertinent to appropriately incorporating DARC into the routine clinical management of patients harboring pre-malignant and pre-invasive lesions. Furthermore, exposure and response evaluation of these model systems to aging-prevention drugs (i.e., metformin, rapamycin) and approved immunotherapeutics based on DARC status may yield critical insight into appropriate therapeutic intervention for high-risk patients. The establishment of optimal cut-offs for DARC IHC expression that is associated with risk of progression and therapeutic response will also be necessary for exploiting DARC expression to guide and inform clinical decision-making. Hence, the investigation into this unique role of DARC could be a potential game-changer in high-risk patient management by (1) better informing clinical decisions, (2) allowing for the avoidance of unnecessary administration of cytotoxic and invasive treatments, and (3) providing targeted, more “cytofriendly” therapeutic approaches, which may ultimately “turn back the clock on time” for inflamm-aging-prone high-risk patients.

## Figures and Tables

**Figure 1 cells-11-03818-f001:**
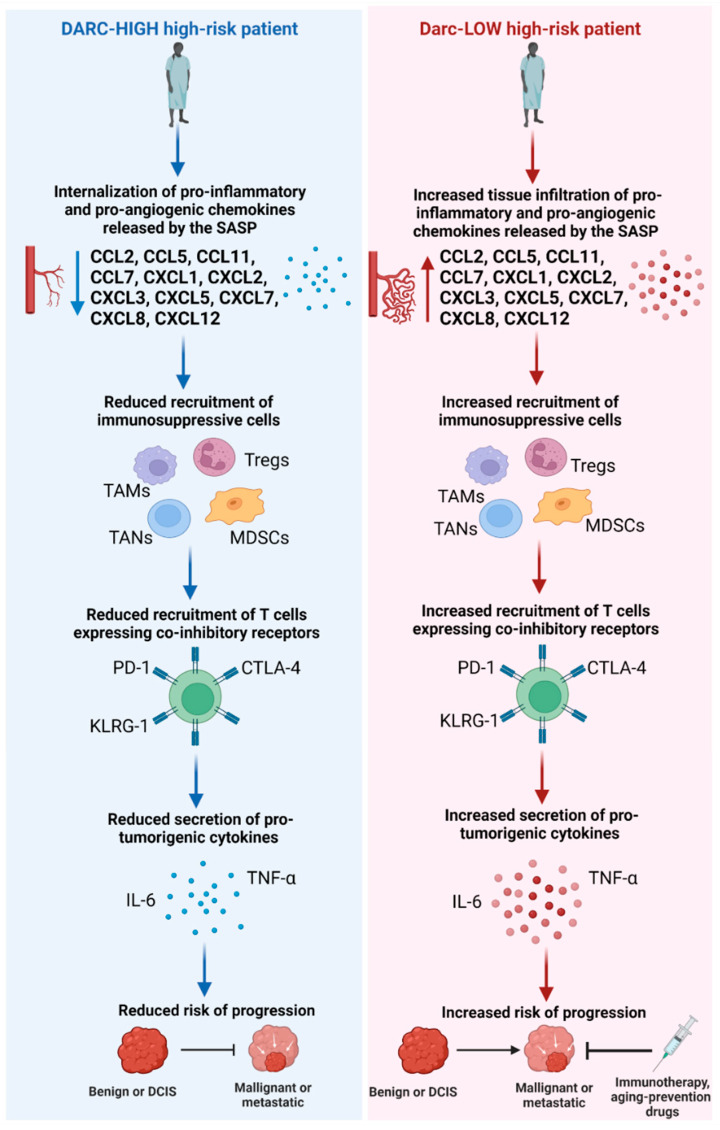
Potential role of DARC expression in influencing risk of inflamm-aging-mediated progression among high-risk patients. Illustration of how endothelial or epithelial DARC IHC expression levels may be influencing risk of progression among patients with high-risk lesions. High-risk patients with low DARC IHC expression may exhibit increased tissue infiltration of pro-tumorigenic cytokines and chemokines, immunosuppressive cells and T cells expressing co-inhibitory molecules, which can increase likelihood for neoplastic transformation and progression to occur. Early-stage immunotherapeutic or age-prevention drug intervention based on DARC status could potentially derail progression to malignancy or metastatic breast cancer.

## Data Availability

Not applicable.

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
