# Peer review of "The DARC Side of Inflamm-Aging: Duffy Antigen Receptor for Chemokines (DARC/ACKR1) as a Potential Biomarker of Aging, Immunosenescence, and Breast Oncogenesis among High-Risk Subpopulations"

_cells, 2022, doi:10.3390/cells11233818_

Round 1

Reviewer 1 Report

Given that the publication is a review, the authors may include a table listing the most recent ageing, immunosenescence, and breast oncogenesis biomarkers.. 

Author Response

Reviewer 1: Given that the publication is a review, the authors may include a table listing the most recent ageing, immunosenescence, and breast oncogenesis biomarkers.. Response: Great suggestion! We have included a table on inflamm-aging-related biomarkers and pathways in Table 1 located in the introduction section (Table 1).

Reviewer 2 Report

This is a comprehensive review article about the role of the Duffy antigene chemokine receptor (DARC/ACKR1) from members of a premier cancer center. Therein the authors postulate that DARC/ACKR1 is critical for the maintenance of the immune homeostasis and as such may prevent breast tumor progression. The underlying hypothesis is that DARC/ACKR1 could be a biomarker for breast oncogenesis. Based on this very interesting premise the authors compiled a well-researched and well-written review article that covers the most important aspects of inflammation in cancer. A point of critique could be that the manuscript is too long and, oftentimes, too general. The lack of references after critical statements makes it difficult to discern proof from conjecture. Moreover, their line of reasoning is too often based on indirect evidence instead of what has been published about DARC/ACKR1 in the context of cancer. Therefore, I believe that the manuscript could be significantly improved if the authors limit the introduction to what’s important about the function of DARC/ACKR1 and then focus on DARC/ACKR1 in health and disease. In addition, they should focus on studies about DARC/ACKR1 in the context of cancer and then build their case from there. They should state more clearly if they write about indirect evidence.

Author Response

This is a comprehensive review article about the role of the Duffy antigene chemokine receptor (DARC/ACKR1) from members of a premier cancer center. Therein the authors postulate that DARC/ACKR1 is critical for the maintenance of the immune homeostasis and as such may prevent breast tumor progression. The underlying hypothesis is that DARC/ACKR1 could be a biomarker for breast oncogenesis. Based on this very interesting premise the authors compiled a well-researched and well-written review article that covers the most important aspects of inflammation in cancer.

Response: Thank you for your kind points!

A point of critique could be that the manuscript is too long and, oftentimes, too general.

Response: Great point. We have removed some content throughout the Introduction, “A light in the DARC,” and Discussion sections, in order to streamline and condense the article, and to make it more cancer-focused.

The lack of references after critical statements makes it difficult to discern proof from conjecture.

Response: We do believe that we have included primary references for all critical statements, and in fact, our manuscript has over 210 references. We would like to highlight that we did remove some sentences from the Introduction that we felt were presented a little too prematurely in the article (i.e., before the multifarious roles of DARC had been discussed in depth), which may have conveyed the impression that they were somewhat conjectural. Additionally, we have rephrased some sentences throughout the article to clarify which statements reflect our perspectives.

Moreover, their line of reasoning is too often based on indirect evidence instead of what has been published about DARC/ACKR1 in the context of cancer. Therefore, I believe that the manuscript could be significantly improved if the authors limit the introduction to what’s important about the function of DARC/ACKR1 and then focus on DARC/ACKR1 in health and disease. In addition, they should focus on studies about DARC/ACKR1 in the context of cancer and then build their case from there. They should state more clearly if they write about indirect evidence.

Response: As mentioned above, we have removed some sentences from the Introduction to enhance the section’s focus on DARC’s functions, particularly in the context of cancer. The idea that DARC could serve as a biomarker of oncogenesis is, as acknowledged by Reviewer #2, a novel one, based upon a body of direct evidence, as well as lines of indirect, but nevertheless highly supportive evidence, from diverse cancer-related studies related to the tumor microenvironment, inflammation, and chemokines. For example, we have included both direct and indirect evidence on specific chemokines that DARC clears in the context of inflammation, that are also reported to be involved in carcinogenesis (in the “Painting a DARC picture” section). We noted some “indirect lines of evidence” in sentences with specific but important conclusions in the “Painting a DARC picture” section. The functions of DARC/ACKR1 are discussed in Introduction and extensively in the “A light in the DARC” section. Studies discussing DARC/ACKR1’s direct role in the context of cancer, and specifically, carcinogenesis, are mentioned in the “A light in the DARC” and “Painting a DARC picture” sections, specifically in paragraph 5 of the “Painting a DARC picture” section. By presenting all relevant lines of evidence from health and disease states, we hope to make a compelling case that DARC/ACKR1 may be a valuable biomarker of oncogenesis, that the field may be under-appreciating and not paying adequate attention to.

Reviewer 3 Report

The article is well written and focused. I Appreciate the authors for this part of work. 

Author Response

Response: Thanks for your kind comments!